# Envelope Proteins of Hepatitis B Virus: Molecular Biology and Involvement in Carcinogenesis

**DOI:** 10.3390/v13061124

**Published:** 2021-06-11

**Authors:** Jun Inoue, Kosuke Sato, Masashi Ninomiya, Atsushi Masamune

**Affiliations:** Division of Gastroenterology, Tohoku University Graduate School of Medicine, 1-1 Seiryo-machi, Aoba-ku, Sendai 980-8574, Japan; ksato9139@gmail.com (K.S.); m-nino5@hello.odn.ne.jp (M.N.); amasamune@med.tohoku.ac.jp (A.M.)

**Keywords:** HBV, envelope, Dane particle, subviral particle

## Abstract

The envelope of hepatitis B virus (HBV), which is required for the entry to hepatocytes, consists of a lipid bilayer derived from hepatocyte and HBV envelope proteins, large/middle/small hepatitis B surface antigen (L/M/SHBs). The mechanisms and host factors for the envelope formation in the hepatocytes are being revealed. HBV-infected hepatocytes release a large amount of subviral particles (SVPs) containing L/M/SHBs that facilitate escape from the immune system. Recently, novel drugs inhibiting the functions of the viral envelope and those inhibiting the release of SVPs have been reported. LHBs that accumulate in ER is considered to promote carcinogenesis and, especially, deletion mutants in the preS1/S2 domain have been reported to be associated with the development of hepatocellular carcinoma (HCC). In this review, we summarize recent reports on the findings regarding the biological characteristics of HBV envelope proteins, their involvement in HCC development and new agents targeting the envelope.

## 1. Introduction

Hepatitis B virus (HBV) infects 292 million people chronically in the world despite the availability of an effective vaccine [1], and they are at risk of liver cirrhosis and hepatocellular carcinoma (HCC) [2]. Although antiviral therapies such as nucleos(t)ide analogues (NAs) and interferon have been developed, complete eradication of HBV is still difficult because covalently closed circular DNA (cccDNA), which is a highly stable and active episomal genome, is formed in the nucleus of HBV-infected hepatocytes [3]. Even if patients have received NA therapy and HBV replication is inhibited, they still have the potential to develop liver cancer [4,5]. Therefore, novel antiviral therapies are required clinically and several lines of clinical trials for HBV chronic infection are ongoing [6,7].

The mature infectious HBV particle with a diameter of 42 nm, which is also called a “Dane particle”, consists of an envelope and nucleocapsid containing HBV DNA and polymerase (Figure 1). A large amount of hepatitis B surface antigen (HBsAg) subviral particles (SVPs), which do not include nucleocapsid, is released from HBV-infected hepatocytes [8] and HBsAg assays detect both Dane particles and SVPs. Additionally, the infected hepatocytes release several kinds of other immature particles [9]. Hepatitis B e antigen (HBeAg) is also secreted from the infected hepatocytes and is detected in the serum of patients. HBeAg seroconversion, defined as loss of HBeAg and appearance of anti-HBe antibody, occurs both in the natural course and under treatment. HBeAg-defective precore mutants are found in most inactive HBV carriers with undetectable HBeAg [10,11]. SVPs and HBeAg are not essential for the infection, but they are considered to help the establishment of persistent infection of HBV via effects on the immune system [8]. The HBV envelope, which is considered to be essential for the infection [12], is composed of a cellular lipid bilayer and three HBV surface proteins: large/middle/small hepatitis B surface proteins (LHBs/MHBs/SHBs). The ratio of LHBs, MHBs and SHBs in the envelope of Dane particles was reported to be approximately 1:1:4 [13]. Moreover, hepatitis D virus (HDV) requires the HBV envelope for the packaging. Therefore, the HBV envelope is indispensable for the life cycle of HBV and HDV and could be an antiviral target against these viruses that are tough to beat.

As for the carcinogenesis of HBV, hepatitis B x protein (HBx) and LHBs are considered to play major roles in the infected hepatocytes. HBx is included often in the HBV DNA that is integrated into the host DNA of HCC tissue and has been reported to have multifunctional roles in the modulation of the expression/activities of many genes [14]. LHBs accumulation in the endoplasmic reticulum (ER) has been documented to associate with ER stress responses, which lead to pro-oncogenic effects [15]. In this review, we summarize the biological characteristics of HBV envelope and SVPs and the roles of envelope proteins in the hepatocarcinogenesis.

## 2. Biological Characteristics of HBV Envelope

The HBV envelope proteins, LHBs, MHBs and SHBs, are translated from 2.4/2.1 kb mRNAs, which are transcribed from cccDNA. Additionally, these mRNAs are transcribed from integrated viral genome. The C-terminus of LHBs, MHBs and SHBs is common, and the length is 400 (389), 281 and 226 amino acids, respectively (Figure 2A). The LHBs consists of preS1, preS2 and S domains, and MHBs consists of preS2 and S domains. There are 4 transmembrane domains in SHBs and “a” determinant that is exposed on the outside of the envelope (Figure 2B). The “a” determinant is an antigenic domain that is a major target of neutralizing antibodies [16]. The N-terminus is myristoylated at glycine 2, which is considered to be essential for the infection [17]. Myristoylation is a posttranslational modification and myristic acid is attached by the ubiquitous enzyme N-myristoyltransferase (NMT). Because the myristic acid is a hydrophobic moiety, the myristoylated part is inserted into hydrophobic regions in the lipid bilayer [18]. Using the myristic acid as an anchor, the preS1 and preS2 in LHBs are considered to be able to transit between the cytoplasmic side and the luminal side (Figure 2B) [19]. The preS1 on the cytoplasmic side is required for the interaction between the LHBs and capsid at the step of envelope formation [20], and the preS1 on the luminal side, which is exposed on the Dane particle, is essential for the infection by interacting with the HBV receptor, sodium-taurocholate cotransporting polypeptide (NTCP) [21].

The three HBV envelope proteins are glycosylated at an N-linked glycosylation site (N-X-S/T) (Figure 2A). All three proteins have a potential N-glycosylation site at N146 of the S domain, but LHBs and SHBs are also present in non-glycosylated forms. MHBs has an additional N-glycosylation site at N4 of the preS2 domain, which is not glycosylated in LHBs, probably because of the altered conformation of preS1/preS2 domains [22]. As a result, LHBs exists in 2 forms (p39 and gp42), MHBs in 2 forms (gp33 and gp36 [a diglycosylated form]) and SHBs in 2 forms (p24 and gp27). Additionally, HBV genotypes C and D have an O-linked glycosylation site at T34 of the preS2 domain [23]. The preS2 glycosylation is central to folding, assembly and secretion [24]. Additionally, N-glycosylation in envelope proteins has dual roles in the interaction with immune systems including dendritic-cell-specific ICAM-3-grabbing nonintegrin (DC-SIGN) [25] and in the occlusion of neutralizing epitopes [26].

A compartment of the late endocytic pathway, the multivesicular body (MVB), which generates exosomes as intraluminal vesicles, is considered to participate in the formation of the HBV envelope, based on the findings that endosomal sorting complex required for transport (ESCRT) is necessary for the release of Dane particles [27,28,29]. The nucleocapsids bud inwards into invaginations of the MVB membrane [30] (Figure 3). Direct interaction between the nucleocapsid and envelope proteins was demonstrated in cell-free binding assays [31,32] and cell-based assays [20], and several host factors have been identified to work in the process of envelope formation. Translated LHBs, MHBs and SHBs are inserted into the ER membrane interacting with chaperones such as calnexin [33], Hsc70 and BiP [34]. LHBs interacts with Hsc and BiP and MHBs interacts with calnexin, but SHBs does not interact with these chaperones. These different interactions might be one of the causes of their different destinies. Moreover, γ2-adaptin interacts with LHBs [35] and is considered to assist the transport of LHBs to MVB. Recently, our group reported that the small GTPase Rab5B is required for the transportation of LHBs from ER to MVBs [36]. On the MVB membrane, α-taxilin, which interacts with LHBs and ESCRT I component TSG101, was shown to be essential for the release of Dane particles [37]. Additionally, an exosome marker, CD63, was shown to work in the incorporation of LHBs in the HBV envelope by us [38]. Nedd4 binds to the late domain-like PPAY motif of core protein and it is considered that ubiquitinated Nedd4 binds to γ2-adaptin resulting in a linkage between LHBs and nucleocapsid [30,39].

The Dane particles are thought to be released from hepatocytes via exocytosis in the same manner as exosomes, but the details have yet to be determined. It was demonstrated that Dane particles are released in a form with preS1 hidden in the interior and that they are converted spontaneously into a form with preS1 on the surface [40]. Because the preS1 domain binds with heparan sulfate proteoglycans on the cellular membrane of various cell types, the dynamic topology switch is thought to be utilized to avoid non-productive attachment before the Dane particles reach the liver of a naïve host. A part of MVBs are considered to be fused with lysosome, resulting degradation of cargos. The small GTPase Rab7, which was activated by HBV infection, was shown to enhance the fusion and to regulate the HBV secretion [41].

## 3. Subviral Particles of HBV

SVPs were found in HBV-infected patients in 2 forms: sphere and filament (Figure 1). The sphere is 22 nm in a diameter, and the filament is variable in length. The secretory pathways of these SVPs were considered to be distinct from Dane particles. A recent report suggested that coat protein complex II (COPII) is required for the transport of SHBs from ER to ER-Golgi intermediate compartment (ERGIC) [42], where filamentous structures are unpacked and relaxed to form SVP spheres [43]. These are considered to be secreted via Golgi. Also, SVP filaments that contain a larger amount of LHBs than spheres were reported to be ESCRT-dependently released via MVBs [44].

## 4. HBV Envelope Proteins as Clinical Diagnostic Tools

As described above, a large amount of particles containing HBV envelope proteins are present in serum of HBV-infected patients and are detected as HBsAg. The serum HBsAg assays using techniques such as chemiluminescence immunoassay (CLIA) or chemiluminescence enzyme immunoassay (CLEIA) are easy methods to distinguish the HBV-infected patients, but the sensitivity is lower than serum HBV DNA in general, and sometimes underdiagnose patients who are at an early stage of infection or reactivation of HBV. To improve the sensitivity, immune complex transfer (ICT)-CLEIA, which includes a sample pretreatment to dissociate HBsAg-antibody immune complexes, has been developed [45], and its application for the detection of HBV reactivation following chemotherapies for hematological malignancies was reported [46,47]. In the late stage of infection, HBsAg is detectable for a long time even in patients whose HBV DNA became undetectable.

HBsAg quantification is considered to be a surrogate marker of viral suppression in patients under NA treatments whose serum HBV DNA is undetectable. HBsAg loss is regarded as the optimal treatment endpoint, termed as “functional cure” [2]. The level of HBsAg, as well as hepatitis B core-related antigen (HBcrAg), is associated with relapse of hepatitis after discontinuation of NA [48]. Moreover, high levels of HBsAg were reported to be a risk factor for HCC in patients with low HBV DNA levels [49].

As a novel evaluation method of HBV envelope proteins, quantification of LHBs and MHBs has been reported to predict clinical outcome [50,51]. Inactive HBV carriers had lower proportion of LHBs and MHBs [50], and during antiviral therapies, the decline of LHBs and MHBs was found before subsequent HBsAg loss [51].

## 5. Genetic Variations of HBV Envelope

Several mutations/deletions are found in HBV-infected patients and associations with specific disease outcomes have been reported. Vaccine-escape mutations such as G145R and D144A/E are variations that were found in the “a” determinant of HBV from patients with a vaccine-induced antibody response [16]. Moreover, these mutations were found in HBV-reactivated patients who were treated with anti-cancer chemotherapies or immunosuppression therapies [52]. Interestingly, several mutations in the “a” determinant that generate novel N-glycosylation sites were reported including M133T, which restores the secretion impairment of virion with G145R [53].

Many previous studies reported that preS1/preS2 deletions were found in patients with chronic HBV infection and were associated with the liver disease progression. Deletion in the preS1 C-terminal half was more frequently found in patients with chronic hepatitis and liver cirrhosis than in asymptomatic carriers [54], and deletion in the preS2 region was frequently found in HCC patients [55]. Especially, preS2-defective mutations including mutations at the level of preS2 start codon and deletions at the 5′-terminal half of the preS2 region were frequently found [56]. The frequency of deletions in these regions was low in patients with acute HBV infection, indicating that these deletions may emerge during the long-term persistence of the viral genome [57]. As described later, preS1/preS2 deletion mutants are reported to be retained in ER and subsequently induce ER stress [58].

A deletion of 11 amino acids at the N-terminus of preS1 domain is found almost specifically in genotype D HBV, and two recent reports showed that this deletion enhances the binding and infectivity to hepatocytes [59,60]. This deletion was found also in HBV strains of nonhuman primates, suggesting that the deletion may have advantages in the HBV life cycle beyond the host species [60].

## 6. HBV Envelope Protein as a Cause of HCC

The preS mutations including deletions in the preS1 and preS2 regions were found frequently in HCC patients [61] and many studies using in vitro and in vivo models showed an association between preS mutations and HCC. An association of preS mutations with HCC development was reported, including patients with low HBV DNA and alanine aminotransferase levels [62]. Because the deletion site of preS2 is known to include an epitope of the T-cell response and B-cell neutralization [63], HBV clones with such deletions might be selected as immune escape mutants [58]. These regions overlap the polymerase gene but correspond to the spacer region, which is not required for the enzyme activity [64]. LHBs with preS1 or preS2 deletion accumulates in ER and LHBs-accumulated hepatocytes are observed as ground-glass hepatocytes [58], which have been recognized as the histological hallmark of chronic HBV infection [65]. The misfolded protein of the preS1/preS2 mutants induces strong and sustained ER stress [15]. ER stress increases intracellular Ca^2+^ level that leads to mitochondrial dysfunction, hepatocyte apoptosis and liver fibrosis, resulting in the promotion of carcinogenesis [62]. The integrated HBV DNA can supply LHBs including its mutants, like as a hepatoma cell line PLC/PRF/5 [66].

The preS2 mutant initiates a mammalian target of rapamycin (mTOR)-dependent glycolytic signal cascade and might promote tumorigenesis by aerobic glycolysis [67]. Moreover, transgenic mouse livers harboring preS2 mutant showed lipid accumulation through the activation of sterol regulatory element binding transcription factor 1 (SREBF1) by mTOR signaling [68]. These changes in metabolic pathways might be linked to the HCC development. Additionally, negative feedback from the regulation of HBsAg synthesis by activation of the mTOR signal pathway was reported [69], indicating that a decreased level of HBsAg and HBV DNA in the serum may not necessarily represent a good sign during the clinical course.

Importantly, antiviral therapy with NAs has been reported to reduce HCC development [70,71], including patients infected with preS-mutant HBV [62]. Because the suppression effects of NAs on serum HBsAg [4] and intracellular cccDNA [72] are not ideal, NAs might not reduce accumulation of LHBs enough. Therefore, novel drugs that can inhibit the accumulation of LHBs in ER might reduce HCC further, which is still found in some patients who are treated with NAs [4,5].

## 7. HBV Envelope as a Target of Antiviral Therapy

### 7.1. Inhibition of HBV Entry

Recently, many studies have reported agents that inhibit the interaction between HBV preS1 and NTCP, including bulevertide (formerly Myrcludex B), a peptide that comprises N-terminal 47 preS1 amino acids with an N-terminal myristoyl moiety, inhibits the HBV and HDV entry [73,74] and was approved for chronic HDV infection. Moreover, cyclosporine A and cyclosporine derivatives [75,76,77], vanitaracin A [78], Ro41-5253 [79] and evans blue [80] were reported to bind to NTCP and inhibit its interaction with preS1. Some of these interfere with the primary NTCP function and inhibit bile acid intake, and selective inhibitors of HBV/HDV infection could be candidates for clinical application. Additionally, preS1-specific neutralizing antibodies induced by DNA vaccine containing preS1 and the large HDV antigen were reported to inhibit HBV entry in an animal model [81]. The HBV entry inhibitors are not expected to deplete cccDNA completely, but de novo infection via NTCP is reported to be necessary to maintain the cccDNA pool [82], and the benefit of entry inhibition might be more significant than expected [7,83].

### 7.2. Inhibition of HBV Envelope Protein Release

Because HBsAg SVPs are considered to induce exhaustion of the adaptive immune responses against HBV [8], the inhibition of HBsAg release can be a target for the HBV suppression. Nucleic acid polymers (NAPs) are single-stranded phosphorothioated oligonucleotides and interact with exposed amphipathic alpha helices in proteins in a sequence-independent manner [84], and some NAPs such as REP 2139 and REP 2165 were shown to inhibit HBsAg release; clinical trials are ongoing [83]. Especially, its combination with pegylated interferon (PEG-IFN) plus tenofovir disoproxil fumarate was reported to show a good response rate for HBsAg loss and HBsAg seroconversion during/after the therapy [85].

NAPs selectively inhibit the secretion of spherical SVPs without affecting Dane particles. Additionally, they reduced also intracellular HBsAg in an in vitro model probably via proteasomal and lysosomal degradation pathways [86].

### 7.3. Inhibition of HBV Enveloping of the HBV Nucleocapsids

As described before, the interaction between preS1 and nucleocapsids is necessary for the formation of infectious HBV particles. Only one report showed agents that inhibited the interaction between the envelope and the nucleocapsid [87], and further investigation for this step is required.

## 8. Conclusions

The envelope of HBV is indispensable for the life cycle of HBV and could be a promising target for anti-viral therapies. Because the accumulation of envelope proteins including the deletion mutants in ER cause ER stress leading to carcinogenesis, suppression of the envelope protein expression and/or reduction of its mutants might be a therapeutic target to decrease HCC patients. Further studies to clarify the biology of the HBV envelope and the development of agents targeting the envelope are required for the establishment of novel antiviral therapy.

## Figures and Tables

**Figure 1 viruses-13-01124-f001:**
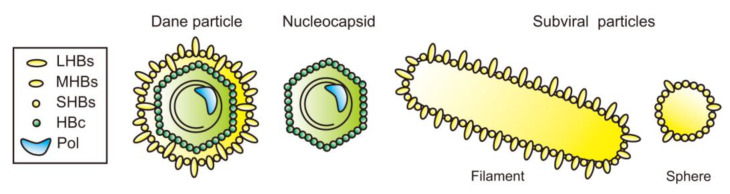
Schema of HBV-associated particles. Dane particles are 42 nm in diameter, and their envelope consists of large/middle/small hepatitis B surface proteins (LHBs/MHBs/SHBs) and a nucleocapsid containing viral genome and polymerase. There are 2 forms of subviral particles, filament and sphere. The former consists of LHBs, MHBs and SHBs, and the latter consists of MHBs and SHBs.

**Figure 2 viruses-13-01124-f002:**
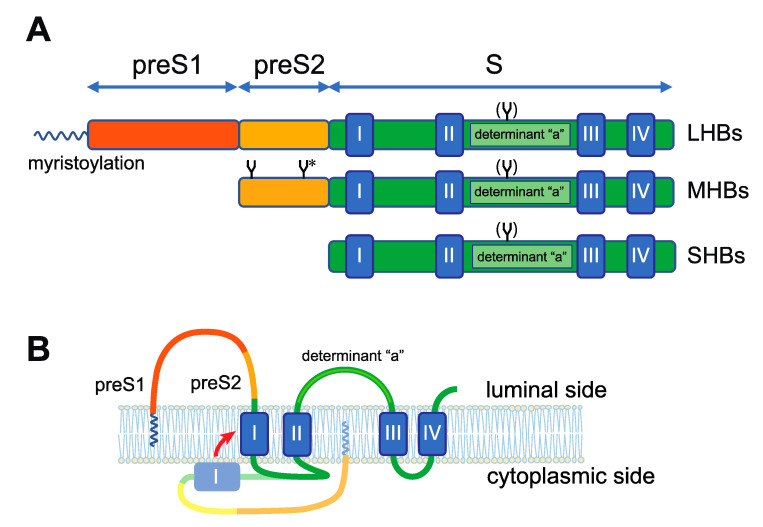
The structure of HBV envelope proteins. (**A**) LHBs contains preS1, preS2 and S domains. Typically, the length is 400 amino acids for most genotypes and 389 amino acids for genotype D. Myristic acid, which is a hydrophobic moiety, is attached at the N-terminus. MHBs contains preS2 and S domains, and SHBs contains only the S domain. There are 4 transmembrane domains (I–IV) in the S domain. Two N-glycosylation sites are indicated as “Y” and the O-glycosylation site for genotypes C and D is indicated as “Y*”. “(Y)” indicates only the part of the proteins that is glycosylated. (**B**) Two-dimensional conformation of LHBs. The myristic acid acts as an anchor and the preS1/preS2 domains in LHBs move between the cytoplasmic side and the luminal side.

**Figure 3 viruses-13-01124-f003:**
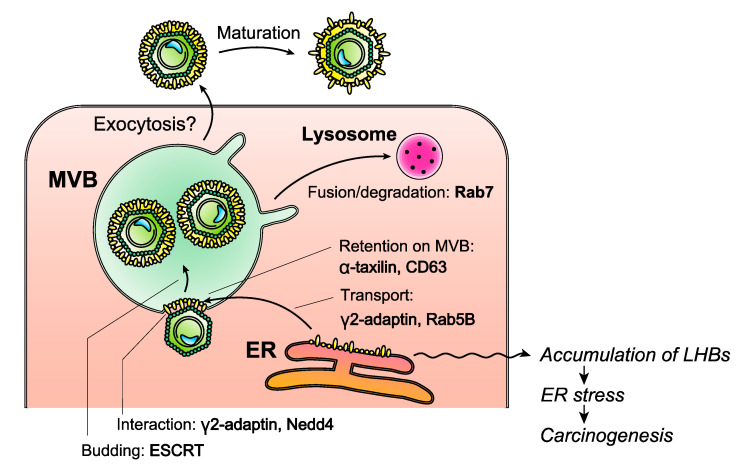
Model depicting the steps of HBV envelope formation. Molecules/organelles in the cells that are utilized for the envelope formation are shown. Nucleocapsid is considered to interact with preS1/preS2 of LHBs on the cytoplasmic side on the multivesicular body (MVB) membrane and buds into the MVB with the action of endosomal sorting complex required for transport (ESCRT). After the release from cells, probably via exocytosis, the position of the preS1/preS2 domains changes to the luminal side to attach to the surface of hepatocytes. LHBs accumulation in the endoplasmic reticulum (ER), which is accelerated by preS mutation, leads to carcinogenesis.

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
