# Peer review of "Envelope Proteins of Hepatitis B Virus: Molecular Biology and Involvement in Carcinogenesis"

_viruses, 2021, doi:10.3390/v13061124_

Round 1
Reviewer 1 Report
Comments
Line 31:“A mature infectious HBV particle” is not appropriate. “The matureinfectious HBV particle” is a better phrase.
Line 33 “A large amount of subviral particles (SVPs), which include no nucleocapsid” would be clearer as “A large amount of HBsAg subviral particles (SVPs), which do not include nucleocapsid”
Line 37 “is secreted also”please change to “is also secreted”
Line 38” but, in many cases, the status of HBeAg converts to negative during the clinical course” This needs to be rephrased. For example, after a full stop “HBeAg seroconversion, defined as loss of HBeAg and appearance of anti-HBe antibody occurs both in the natural course and under treatment”
Line 40 “without HBeAg” means undetectable HBeAg?
Line 62. There is no mention of HBsAg transcription from integrated HBV sequences. The relative amount of L, M and S proteins of the envelope is not discussed.
Line 147 HBV Envelope Proteins as Clinical Diagnostic Tools: HBsAg detection is more sensitive than HBV DNA in HBeAg negative chronic infection (inactive carrier state). Moreover, quantitative HBsAg is not discussed.
Line 187. In this section the effect of HBsAg integrated sequences could be added.
Author Response
We appreciate the helpful comments that helped to improve the manuscript. We have taken all suggestions to heart through editorial additions. We hope these additions are sufficient but can make additional alterations if need be. A point by point response is below.
Comment: Line 31:“A mature infectious HBV particle” is not appropriate. “The mature infectious HBV particle” is a better phrase.
Response: Thank you for the comment. We corrected it.
Comment: Line 33 “A large amount of subviral particles (SVPs), which include no nucleocapsid” would be clearer as “A large amount of HBsAg subviral particles (SVPs), which do not include nucleocapsid”
Response: According to the comment, we modified the part.
Comment: Line 37 “is secreted also”please change to “is also secreted”
Response: We corrected it.
Comment: Line 38” but, in many cases, the status of HBeAg converts to negative during the clinical course” This needs to be rephrased. For example, after a full stop “HBeAg seroconversion, defined as loss of HBeAg and appearance of anti-HBe antibody occurs both in the natural course and under treatment”
Response: We modified the part.
Comment: Line 40 “without HBeAg” means undetectable HBeAg?
Response: Yes, it is. We change this part to “with undetectable HBeAg”.
Comment: Line 62. There is no mention of HBsAg transcription from integrated HBV sequences. The relative amount of L, M and S proteins of the envelope is not discussed.
Response: We agree with these comments. We added sentences “Additionally, these mRNAs are transcribed from integrated viral genome” and “The ratio of LHBs, MHBs and SHBs in the envelope of Dane particles were reported to be approximately 1:1:4”.
Comment: Line 147 HBV Envelope Proteins as Clinical Diagnostic Tools: HBsAg detection is more sensitive than HBV DNA in HBeAg negative chronic infection (inactive carrier state). Moreover, quantitative HBsAg is not discussed.
Response: Thank you for the comment. These are important points clinically. We added sentences ”In the late stage of infection, HBsAg is detectable for a long time even in patients whose HBV DNA became undetectable” and “HBsAg quantification is considered to be a surrogate marker of viral suppression in patients under NA treatments whose serum HBV DNA is undetectable. HBsAg loss is regarded as the optimal treatment endpoint, termed as “functional cure”. The level of HBsAg, as well as hepatitis B core-related antigen (HBcrAg), is associated with relapse of hepatitis after discontinuation of NA. Also, high levels of HBsAg were reported to be a risk factor for HCC in patients with low HBV DNA levels”.
Line 187. In this section the effect of HBsAg integrated sequences could be added.
Response: We added a sentence to this section as “The integrated HBV DNA can supply LHBs including its mutants, like as a hepatoma cell line PLC/PRF/5”.
Reviewer 2 Report
Review on Viru-1255288
Authors described general consideration about hepatitis B virus (HBV) envelope proteins. The context seems not to be new and not so exiting. The authors, however, described about it with good balance.
Specific points;
- Line 33-36, the author described “the infected hepatocytes release several kinds of other immature particles including naked nucleocapsid”. This description/expression is very unclear and confusing. Ref 9 never says that naked nucleocapsid is released/secreted. It is true that the naked nucleocapsid is released from the damaged hepatocytes infected with HBV. If the author insists the release/secretion of naked nucleocapsids, cite a appropriate and confirmable publication as a reference other than the ref 9. And the authors should distinguish the release from the secretion.
- Line 63, “N-terminus of” should be “C-terminus of”.
- Line 242, the title of the section “Inhibition of HBV Envelope Formation” is very confusing and unclear. This section describes enveloping of the HBV capsid. Then the title should be “Inhibition of HBV Enveloping of the HBV nucleocapsids”.
- The authors say that accumulation of LHBs is considered to promote hepatocarcinogenesis and described “Inhibition of HBV Envelope Protein Release” in the section 7.2. Thus, inhibitors of HBV envelope protein release must be reconsidered in terms of hepatocarcinogenesis. Actually, how do these kinds of drugs make effects on liver cell growth, transformation and so on. The authors should discuss about this point while citing proper publications.
Author Response
We appreciate the helpful comments that helped to improve the manuscript. We have taken all suggestions to heart through editorial additions. We hope these additions are sufficient but can make additional alterations if need be. A point by point response is below.
Comment: Line 33-36, the author described “the infected hepatocytes release several kinds of other immature particles including naked nucleocapsid”. This description/expression is very unclear and confusing. Ref 9 never says that naked nucleocapsid is released/secreted. It is true that the naked nucleocapsid is released from the damaged hepatocytes infected with HBV. If the author insists the release/secretion of naked nucleocapsids, cite a appropriate and confirmable publication as a reference other than the ref 9. And the authors should distinguish the release from the secretion.
Response: Thank you for the comment. The naked nucleocapsid has been described in only in-vitro studies. This point is not essential in this paper and we removed “including the naked nucleocapsid”.
Comment: Line 63, “N-terminus of” should be “C-terminus of”.
Response: We are sorry for the mistake. We corrected it.
Comment: Line 242, the title of the section “Inhibition of HBV Envelope Formation” is very confusing and unclear. This section describes enveloping of the HBV capsid. Then the title should be “Inhibition of HBV Enveloping of the HBV nucleocapsids”.
Response: Thank you for the suggestion. We modified the section title according to the comment.
Comment: The authors say that accumulation of LHBs is considered to promote hepatocarcinogenesis and described “Inhibition of HBV Envelope Protein Release” in the section 7.2. Thus, inhibitors of HBV envelope protein release must be reconsidered in terms of hepatocarcinogenesis. Actually, how do these kinds of drugs make effects on liver cell growth, transformation and so on. The authors should discuss about this point while citing proper publications.
Response: Thank you for the constructive comment. As you pointed out, accumulation of LHBs by inhibitors of HBV envelope protein have a potential to cause ER stress, but NAP decreased intracellular HBsAg as well as secreted HBsAg. This may be due to proteasomal and lysosomal pathway (Boulon R et al. Antiviral Res 2020). We added a description on this point as “NAPs selectively inhibit the secretion of spherical SVPs without affecting Dane particles. Additionally, they reduced also intracellular HBsAg in an in-vitro model probably via proteasomal and lysosomal degradation pathways”.